# Dissecting Polygenic Etiology of Ischemic Stroke in the Era of Precision Medicine

**DOI:** 10.3390/jcm11205980

**Published:** 2022-10-11

**Authors:** Jiang Li, Vida Abedi, Ramin Zand

**Affiliations:** 1Department of Molecular and Functional Genomics, Weis Center for Research, Geisinger Health System, Danville, PA 17822, USA; 2Department of Public Health Sciences, College of Medicine, The Pennsylvania State University, Hershey, PA 17033, USA; 3Department of Neurology, College of Medicine, The Pennsylvania State University, Hershey, PA 17033, USA; 4Neuroscience Institute, Geisinger Health System, 100 North Academy Avenue, Danville, PA 17822, USA

**Keywords:** genome-wide association study, ischemic stroke, stroke subtypes, cerebral small vessel disease, polygenic risk score, mendelian randomization, machine learning, artificial intelligence, electronic health records, gene ontology, least absolute shrinkage and selection operator (LASSO), all-cause mortality, survival analysis, cox proportional hazards regression

## Abstract

Ischemic stroke (IS), the leading cause of death and disability worldwide, is caused by many modifiable and non-modifiable risk factors. This complex disease is also known for its multiple etiologies with moderate heritability. Polygenic risk scores (PRSs), which have been used to establish a common genetic basis for IS, may contribute to IS risk stratification for disease/outcome prediction and personalized management. Statistical modeling and machine learning algorithms have contributed significantly to this field. For instance, multiple algorithms have been successfully applied to PRS construction and integration of genetic and non-genetic features for outcome prediction to aid in risk stratification for personalized management and prevention measures. PRS derived from variants with effect size estimated based on the summary statistics of a specific subtype shows a stronger association with the matched subtype. The disruption of the extracellular matrix and amyloidosis account for the pathogenesis of cerebral small vessel disease (CSVD). Pathway-specific PRS analyses confirm known and identify novel etiologies related to IS. Some of these specific PRSs (e.g., derived from endothelial cell apoptosis pathway) individually contribute to post-IS mortality and, together with clinical risk factors, better predict post-IS mortality. In this review, we summarize the genetic basis of IS, emphasizing the application of methodologies and algorithms used to construct PRSs and integrate genetics into risk models.

## 1. Polygenic Nature of Ischemic Stroke

Ischemic stroke (IS) is a highly complex and heterogeneous disorder caused by multiple etiologies with moderate heritability. Monogenic forms of IS are rare. Some studies have reported 30% to 40% phenotypic variability explained by common genetic variation [1]. All main classification methods stratify IS subtypes into the five major categories: large artery atherosclerosis (LAS), cardiac embolism (CES), small artery occlusion (SVS), uncommon causes, and undetermined causes [2]. The focus of this article is to dissect the etiology of IS through pathway analyses and highlight how statistical methods and machine learning algorithms have contributed to the integration of genetic information into risk models. A flow chart summarizing the topics covered by this review to guide the reader is presented in Figure 1. We first briefly review the genetic basis of monogenic stroke and then turn our attention to the polygenic nature of sporadic IS using polygenic risk scores (PRSs) derived from large-scale genome-wide association studies (GWAS) and meta-analyses of GWAS as a tool to establish a common genetic basis for IS. We will discuss how the polygenic risk for cardiovascular disease may also contribute to the risk for sporadic IS. We will show how PRS may augment IS subtyping and review the polygenic basis of IS subtypes, such as cardioembolic stroke, cerebral small vessel disease (CSVD), and cerebral vascular amyloidosis. Our main focus is pathway-specific PRS analyses. We will show how this approach can leverage information to confirm known and identify novel etiologies related to IS. Some of these specific PRSs may contribute to IS risk stratification for disease/outcome prediction and personalized management. Finally, we will discuss the challenges of integrating PRS into clinical decision support systems and risk stratification procedures.

### 1.1. Pioneer Studies on Monogenetic Disease

Genetic studies contribute significantly to our understanding of the causality of IS and its subtypes. With reference to previous linkage studies, several distinct single-gene variants have been discovered among patients with lacunar stroke and CSVD. CSVD is a common cause of stroke and cognitive impairment in the elderly and affects small vessels of the brain, including small arteries, arterioles, capillaries, and small veins. So-called monogenic cerebrovascular diseases include: (1) cerebral autosomal dominant arteriopathy with subcortical infarcts and leukoencephalopathy (CADASIL), which is the most prevalent monogenic CSVD and is caused by a cysteine-altering mutation in one of the 34 epidermal growth factor-like repeat (EGFr) domains of *NOTCH3* gene at 19q1 [3,4,5]; (2) cerebral autosomal recessive arteriopathy with subcortical infarcts and leukoencephalopathy (CARASIL), which is caused by missense mutations in *HTRA1*, encoding a serine protease, located at 10q26.13 [6]; (3) Fabry disease (FD), a rare X-linked inborn error of glycosphingolipid metabolism resulting from reduced production of lysosomal *α*-galactosidase A (*α*-Gal A), resulting in the accumulation of glycosphingolipids [7] in various cellular compartments, causing structural damage and cellular dysfunction and triggering a secondary inflammatory response, resulting in progressive organ dysfunction [8]; (4) retinal vasculopathy with cerebral leukodystrophy, an autosomal dominant disorder caused by C-terminal frameshift mutations in the Three Prime Repair Exonuclease 1 (*TREX1*) gene located at 3p21.31 [9]; (5) *COL4A1/COL4A2*-related angiopathies; *COL4A1/A2*, located at 13q34, encodes the most abundant and prevalent protein in the basement membrane of all tissues, including cerebral vasculature; type IV collagen helps the basement membrane interact with other cells, playing a role in cell migration, proliferation, differentiation, and survival; and (6) hereditary cerebral amyloid angiopathy (CAA), characterized by cerebrovascular amyloid deposition, mainly observed in leptomeningeal and cortical vessels; it can be classified based on accumulated amyloid proteins, such as amyloid β (*APP*), cystatin C (*CST2*), integral membrane protein 2B (*ITM2B*), prion protein, transthyretin (*TTR*), and others [10].

Understanding the genetics of monogenic CSVD and lacunar stroke [11] can lead to precise diagnosis and prognosis, aid in the development of a targeted treatment plan, and ultimately lead to an improved phenotype definition. Monogenic diseases are rare, and the causal variants have a minor allele frequency (MAF) of less than 0.005 (ultra-rare) in the stroke population. Sporadic IS, which dominates the disease population, cannot be explained by these rare inheritances despite some success in identifying common risk loci at the gene level (e.g., *COL4A2* and *HTRA1*) by the GWAS [11,12,13,14,15].

### 1.2. Genome-Wide Association Studies (GWAS)

Large-scale GWAS on IS and its etiological subtypes have gained popularity in recent years, and some common variants and genes have been identified in the process. A large-scale, multi-ancestry GWAS meta-analysis led by the MEGASTROKE consortium, discovered 22 new stroke risk loci, bringing the total number to 32 loci associated with stroke and stroke subtypes [16]. These genetic variants/genes are mostly subtype-specific, and their biological relevance to the etiology of stroke needs to be further investigated to determine their causality in general, as well as a specific subtype [17]. The variation of effect size observed in different stroke subtypes also poses a challenge for risk stratification.

### 1.3. Polygenic Risk Score (PRS) Construction

PRSs have been used to establish a common genetic basis for complex traits, independent of whether single markers with a significant association or not, and contribute to risk stratification for disease/outcome prediction and personalized management. Although an increasing number of stroke risk loci have been identified, their effect sizes are quite small. [16] In most diseases with a polygenic etiology, genome-wide significant markers explain a small proportion of the heritability of complex traits. However, converging evidence supports the notion that a considerable proportion of phenotypic variation can be explained by the ensemble of individual markers not achieving that level of significance.

A genetic risk score (GRS) derived from multiple loci with genome-wide significant association has a limited power to predict IS [18,19] or its subtypes [20], possibly due to the relatively few identified stroke susceptibility loci and the genetic heterogeneity of IS patients. A GRS from genome-wide loci was shown to be superior to a multi-locus GRS in the prediction of IS in a small Japanese population [21].

To improve the predictive power, the GRS was been extended to loci with small effects and without significant genome-wide association in the form of PRS [22]. There are many ways to construct a PRS. These variations are mainly the result of which genetic variants are included and how their effect sizes are estimated. Thus, PRS can be calculated directly based on individual-level genotyping data or indirectly according to the summary statistics of an ancestry-matched reference panel. Whichever implementation approach is applied for the construction of PRS, the weight of each variant has to be estimated based on a single or multiple large-scale GWAS. These variants can be integrated into a single score by statistical modeling/machine learning approaches, a field that is still growing.

PRSice-2, [23] a *p*-value selection threshold approach with clumping and thresholding (C + T) to filter out SNP in linkage disequilibrium (LD), is a classical way to construct PRSs when assuming a uniform distribution of SNP effect size. PRS_avg_ [24] is constructed by PRSice-2 [23], with the algorithm $PRS_{j}=\underset{i}{\sum }\frac{X_{ij}{\hat{\beta }}_{i}}{m_{j}}$, which is calculated according to the number of observed effective alleles ($X_{ij}$) for each variant multiplied by the corresponding effect size (${\hat{\beta }}_{i})$ derived from MEGASTROKE, divided by the number of alleles ($m_{j}$) included in the PRS from that individual, and the sum of all alleles from that individual (*j*).

The emerging methods could allow the user to specify different heritability models [25]. Because a universal reduction in the effect size of all SNPs could generate a biased PRS, a Bayesian approach to shrinkage of the effect estimates of all SNPs can be applied by calculating a posterior mean effect for each variant based on a prior and subsequent shrinkage based on the extent to which this variant is correlated with similarly associated variants in the reference population. PRS was calculated with the linkage disequilibrium pred algorithm (LDpred) [26], which is adjusted for LD between markers and further rescales allelic effects based on the likelihood of each marker belonging to the fraction of markers assumed to be causal. The prior has two parameters: the heritability explained by the genotypes and the fraction of causal markers with non-zero effects. The heritability parameter is estimated based on GWAS summary statistics and accounts for sampling noise and LD (approximated by a reference panel). Because the causal variant fraction (ρ) is unknown for any given disease, a range of ρ values (i.e., 1, 0.3, 0.1, 0.03, 0.01, 0.003, and 0.001) is typically tested during PRS construction. This process helps to determine which fraction of causal variants produces the best prediction of phenotypes. The PRS calculated by LDpred based on the non-infinitesimal distribution (Gaussian mixture prior) of effect sizes is compared to that from the infinitesimal distribution of effect sizes. As expected, non-infinitesimal modeling of effect size can improve polygenic prediction with larger odd ratios (ORs) than those obtained through infinitesimal modeling (Figure 2), suggesting that only a small fraction of variants are causal.

PRS-CS [27] utilizes a high-dimensional Bayesian regression framework for multivariate modeling of local LD patterns by placing a continuous shrinkage (CS) prior to SNP effect sizes. The user must specify a global shrinkage parameter (ϕ) that reflects the proportion of causal variants, although the program can estimate ϕ based on GWAS results. This method outperforms some existing methods across a wide range of genetic architectures. 

Other methods of PRS construction are varied by the prior assumption of the distribution of SNP effect sizes estimated in joint models that account for LD [25]. Methods also vary in terms of whether they use individual-level data or summary statistics. The methods taking summary statistics into consideration are more appealing, as many institutions share only their summary statistics for collaborative studies to protect patient confidentiality and reduce the risk of data leakage and deanonymization. Lassosum is a method used to compute LASSO/elastic net estimates of a linear regression given summary statistics from GWAS and genome-wide meta-analyses, accounting for LD based on a reference panel [28]. Users often choose multiple methods for PRS construction to determine the robustness and consistency of PRS in association/prediction studies. As a non-modifiable variable, PRS does not provide a baseline or change with the progression of a disease/phenotype. PRS can only explain the relative risk for a disease.

Through a risk stratification by PRS derived from a large-scale GWAS by MEGASTROKE, a recent study showed that the risk of incident stroke in the UK Biobank (UKB) cohort is 35% higher among those in the top third of PRS; this association is independent of lifestyle factors [29]. Genetic overlaps between stroke risk, early neurological changes, and some cardiovascular risk factors (diabetes and hypertension) have also been identified [30].

### 1.4. Low-Frequency Variants Explain More Phenotypic Variation

Previously identified IS risk loci with significant genome-wide association are enriched with low-frequency variants [31]. The partition of SNPs by MAF can provide deep insight into the mechanisms of heritability. If a genetic variant is associated with fitness, selection would drive one allele to low frequency [32]. The latter is the case even for traits without any obvious connection to fitness. The functional architecture of low-frequency variants (0.5% < MAF < 5%) highlights the strength of negative selection across coding and non-coding variants; this effect is also obvious with respect to many cardiometabolic traits [33]. Low-frequency variants bridge the gap between rare variants with putatively larger effect sizes and common variants with smaller effect sizes. Because the loci for cardiovascular diseases are significantly enriched for lifetime reproductive success by natural selection [34] and identified IS subtype-specific loci are more likely to be low in MAF [24,31], we propose that genetic variants with lower MAF may contribute more to the phenotypic variation in IS. When we partitioned the variants by MAF ≤ 0.01, 0.05, 0.1, 0.2, or to all, PRS_LAS_, PRS_CES_, and PRS_SVS_ derived from low-frequency common variants (0.01 < MAF < 0.05) provided the best-fit modeling for our IS cohort (Figure 3), suggesting that low-frequency common variants, when taken together, could contribute more to the risk for matched IS subtypes.

## 2. Polygenic Risk for Cardiovascular Disease May Also Contribute to the Risk for Sporadic IS

Genetic risk factors for IS and its subtypes demonstrate pleiotropism, manifested by varied genetic correlations with many modifiable vascular risk factors, such as hypertension, atrial fibrillation, dyslipidemia, peripheral vascular diseases, diabetes, and perhaps other comorbidities promoting hypercoagulability, such as autoimmune diseases and DVT.

### 2.1. Candidate Gene Approach

Genetic variants from cardiometabolic risk genes directly or indirectly contribute to IS. These candidate genes can be categorized into multiple biological pathways, including the coagulation system (*F5, F2, FGA, FGB, F7, F13A1, VWF,* and *F12*), the fibrinolytic system (*SERPINE1*), platelet receptors (*ITGB3, ITGA2B, ITGA2,* and *GP1BA*), the renin-angiotensin-aldosterone system (*ACE and AGT*), homocysteine and eNOS metabolism (*NOS3, MTHFR, CBS,* and *MTR*), and lipoprotein metabolism (*APOE2, APOE3, APOE4, LPL,* and *PON1*), among others (*PDE4D and ALOX5AP*), which have been reviewed by others [35]. The angiotensin-converting enzyme (*ACE*) deletion allele is associated with increased risk for IS, particularly lacunar stroke [36,37].

### 2.2. Genetic Correlation between Cardiometabolic Risk Factors and IS

A MEGASTROKE analysis identified several genes predisposing to conventional cardiovascular disease, which also increases the risk for stroke [16]. Based on PRS analysis, genetic variants predisposing to thrombosis have been associated with an increased risk of thromboembolic stroke secondary to large artery disease and cardiac embolism, although this genetic predisposition does not apply to small artery stroke, suggesting that thrombotic factors may be less important for CSVD.

Interestingly, shared polygenic risk between small artery stroke and cardiovascular traits, such as systolic/diastolic blood pressure, HDL (inverse correlation), type 2 diabetes (T2D), and coronary artery disease (CAD), has been identified by either or both PRS association study and linkage disequilibrium score regression (LDSC) [16], suggesting that the management of cardiovascular risk factors is important to reduce risk for CSVD in lieu of genetically-based precision medicine.

The shared genetic architecture between white matter hyperintensity (WMH), a surrogate imaging marker for CSVD, and cardiometabolic traits has been evaluated by genome-wide (LDSC [38]) and regional (GWAS-PW [39] and HESS [40]) methods in a large-scale GWAS on WMH volume [15]. A significant genetic correlation has been identified between high WMH volume and high SBP, DBP, SMKindex (lifetime smoking index), body mass index (BMI), and increased risk of venous thromboembolism (VTE). Some genomic regions harbor shared genetic risk variants with at least one other vascular trait, predominantly BP traits, but also BMI, lipid levels, and SMKindex.

### 2.3. Mendelian Randomization for Causal Inference

Genetic correlation between phenotypes (exposure and outcome variable) does not necessarily imply a causal relation. To confirm the causality to the outcome of interest beyond a simple association, causal inference has to be determined according to the Bradford Hill criteria [41]. The causal relationship identified by randomized controlled trials in perspective studies can be simulated via a genetic approach using mendelian randomization (MR) in observational studies.

MR is a method that involves the use of measured variation in known genes to examine the causal effect of a modifiable exposure on disease in observational studies. It is an efficient way to determine causal inference by controlling the known or unknown confounding effects in observational studies. The purpose of MR is to treat genotypes as instrumental variables and to infer the causal effect of an exposure on an outcome variable based on a critical assumption of “instrument strength independent of direct effect” or, alternatively, no direct effect from the genotype to the outcome variable (no pleiotropy) [42]. Three assumptions should be assessed in MR analysis [43]: (1) relevance assumption: the genetic variants associated with the risk factor of interest; (2) independence assumption: there are no unmeasured confounders of the associations between genetic variants and outcome; (3) exclusion restriction: the genetic variants affect the outcome only through their effect on the risk factor of interest.

In an MR study on blood lipids, a genetically elevated level of HDL cholesterol was associated with a reduced risk of small vessel stroke but not with that of large artery stroke and cardioembolic stroke [44]. In the same study, a genetically elevated level of LDL cholesterol was associated with risk of large artery stroke but not small vessel stroke and cardioembolic stroke. Therefore, the genetic risk factors for cardiovascular disease can have differential effects on etiological stroke subtypes [45]. In a large multi-ancestry meta-analysis of GWAS on WMH volume [15], the robustness of MR was demonstrated to confirm the causal association of increasing WMH volume with stroke and Alzheimer-type dementia, as well as of increasing blood pressure with high WMH volume. Similarly, the positive association between cardiovascular risk factors (diastolic, systolic, and pulse pressure; type 2 diabetes; and ever smoking) and lacunar stroke was confirmed [11]. No evidence of pleiotropy was observed in this study, as assessed by MR-Egger without a significant non-zero intercept. There was no evidence of an association with BMI, LDL, or triglycerides.

## 3. Polygenic Risk Scores (PRSs) Augment IS Subtyping

PRSs derived from stroke subtypes may augment the predictive power for patients with a similar etiology. PRSs for atrial fibrillation can significantly explain cardioembolic stroke (CES) risk, independent of other clinical risk factors [46].

We previously showed that PRS_LAS_, PRS_CES_, and PRS_SVS_, which were constructed by the variants with effect size estimated according the MEGASTROKE IS subtypes (LAS, CES, or SVS), explained the most variance of the corresponding subtypes of IS among MEGASTROKE subtypes (larger and warmer dots for the significant level and Nagelkerke pseudo-R^2^, respectively, as indicated by the arrow in Figure 4 using variants from the base file with *p* < 0.1). To determine the robustness of this subtype-specific PRS, a synthesized group (ASL) with more LAS cases (*n* = 120) than SVS cases (*n* = 70) was created. We observed that the predictive power (*R*^2^) and significance was the highest using PRS_svs_, suggesting that there is a lack of a clear boundary between LAS and SVS. However, PRS_CES_ differentiated LAS from CES and SVS from CES (yellow arrows), suggesting that CES has a unique polygenic architecture that separates it from other subtypes. Furthermore, none of the PRSs could significantly explain the phenotypic variation of our ‘Undetermined’ subtype. In summary, some clinical IS subtypes may have distinct or shared polygenic architecture. The effect sizes from low-frequency variants estimated by the summary statistics of GWAS on clinical subtypes contribute more to the polygenic inheritance of the matched subtype.

## 4. Genetic Basis of Cardioembolic Stroke

Atrial fibrillation (AFib) is a leading and independent risk factor for the cardioembolic subtype (CES) of ischemic stroke [47,48]. AFib may be underdiagnosed due to paroxysmal episodes and its asymptomatic nature. Prophylactic anticoagulation therapy remains the standard for IS prevention and outcome improvement for AFib patients. However, AFib undertreatment is a long-existing healthcare concern. Identifying patients at high risk for AFib and subsequent IS could aid in adherence to anticoagulant guidelines [49]. Both AFib and CES are complex disorders with a polygenic nature and top risk loci associated with both disorders are overlapping. For example, the top loci at 4q25 near *PITX2* and 16q22 near *ZFHX3* are also the top risk loci for AFib [50,51,52]. A strong genetic correlation was identified between a previous genetic study of AFib [53] and AFib in the presence of CES [46] in a cohort of the NINDS-Stroke Genetics Network (SiGN), with a Pearson r = 0.76 across SNPs with *p* < 4.4 × 10^−4^ [53]. PRS for AFib is an important independent determinant of overall CES risk after adjustment for clinical AFib risk factors, with OR = 1.40, *p* = 1.45 × 10^−48^ AFib PRS was associated with stroke of undetermined cause (OR = 1.07, *p* = 0.004) but not with other primary stroke subtypes, suggesting the specificity of this AFib PRS for CES [46].

A study focusing on patients with cardiometabolic disease from five large clinical trials showed the power of multi-loci GRS derived from 32-SNP as an independent predictor of IS [54]. All these 32 SNPs were reported as genome-wide association with IS by MEGASTROKE [16]. The predictive value of this GRS appeared the strongest in subjects without previous stroke and subjects with fewer clinical risk factors. Moreover, in patients with AFib but lower CHA_2_DS_2_-VASc scores, the GRS identified patients with risk comparable to those with higher CHA_2_DS_2_-VASc scores, suggesting that this GRS could help to identify patients with AFib but with lower CHA_2_DS_2_-VASc scores in whom high genetic risk might inform decisions about initiating anticoagulation.

A recent study of the UK Biobank Prevalent cohort showed that combining a clinical risk tool (CHA_2_DS_2_-VASc scores) and PRS constructed by lassosum method [28] using summary statistics derived from MEGASTROKE significantly improved the predictability of IS patients with AFib relative to the clinical risk tool alone [55]. This PRS showed no correlation with clinical risk factors and is an independent predictor, with HR = 1.13, 95%CI: 1.06–1.23.

## 5. Genetic Basis of Sporadic Cerebral Small Vessel Disease (CSVD)

CSVD is a syndrome with specific clinical and neuroimaging characteristics, such as WMH, widened perivascular spaces, lacunar infarcts, micro/macro cerebral hemorrhage, and brain atrophy. Heterogeneity in the pathology of CSVD challenge presents with a uniform definition. CSVD can be etiologically classified into six subtypes [56], including (1) arteriolosclerosis (or age- and vascular risk factor-related SVD); (2) cerebral amyloid angiopathy (sporadic and hereditary); (3) inherited SVD (distinct from cerebral amyloid angiopathy (e.g., CADASIL, CARASIL, Fabry disease, and *COL4A1*/*A2* mutations, among others); (4) inflammatory and immune-mediated SVD (systemic and cerebral vasculitis, central nervous system vasculitis, and central nervous system vasculitis secondary to infection); (5) venous collagenosis; and (6) other SVDs (e.g., post-radiation angiopathy and non-amyloid microvascular degeneration in Alzheimer’s disease). The general neuropathological presentation includes focal atherosclerosis at the small perforating arteries, diffused deposition of fibro-hyaline material, and loss of smooth muscle cells in the tunica media with fibrinoid necrosis [57,58,59]. All these pathological changes result in the thickening of the vessel wall and increased blood–brain barrier (BBB) permeability, evidenced by the presence of plasma proteins, such as fibrinogen in the brain parenchyma and cerebrospinal fluid, as well as the leakage of contrast agents across the BBB by MRI.

### 5.1. Heritability of CSVD

CSVD patients may share vascular disease-related risk factors. However, not all cases with known risk factors for vascular diseases present with the clinical symptoms of CSVD; CSVD can be observed in the absence of vascular disease-related risk factors [60]. Thus, the mechanisms underlying these pathological changes cannot be fully explained by the clinical risk factors or using traditional statistical methods.

The most commonly used (endo)phenotype to investigate the heritability of CSVD is the quantitative trait derived from WMH [61]. Heritability for WMH has been estimated to be 55% to 80% based on twin and family studies [62,63,64,65], suggesting that a moderate to a large proportion of variation in WMH can be attributed to genetic effects. Based on GWAS studies, the heritability contributed by common variants has been estimated to be between 13% and 45% [66]. The discrepancy in heritability can be at least partially explained by the contribution of rare variants with larger effect sizes and high penetrance in familial monogenic CSVD versus the heritability contributed by common variants with smaller effect sizes and polygenic inheritance. Several genetic loci have been identified for sporadic CSVD, which are listed in Table 1. However, functional validation should be conducted to determine their exact role in the pathogenesis of CSVD.

### 5.2. Genetic Variants from Extracellular Matrix (ECM) Genes May Contribute to the Risk for Sporadic CSVD

It is possible that ECM genes, such as *COL4A1/A2 and NOTCH3*, mentioned earlier with respect to monogenic CSVD, are involved in the pathogenesis of sporadic CSVD. In a candidate gene study of 888 stroke patients and dementia-free controls, four common variants in *NOTCH3* in high LD with each other were associated with the presence and progression of WMH only in patients presenting with hypertension [78]. These findings were also replicated in the CHARGE cohort (*n* = 8545). However, the association with WMH was not replicated in a meta-analysis of GWAS datasets from IS cohorts in 3670 cases and 7397 controls [78]. All these findings suggest that genetic variation in known monogenic causes of CSVD may contribute to increased risk in a subset of sporadic CSVD.

Common variants from *COL4A1*/*A2* may also contribute to the risk for sporadic CSVD and intracerebral hemorrhage (ICH). A candidate gene-based meta-analysis of GWAS in a subtype of stroke patients and controls with European ancestry identified three common variants, rs9521732, rs9521733, and rs9515199, from the intronic regions of *COL4A2* that were significantly associated with deep ICH. They have a moderate significance in association with lacunar stroke and WMH in symptomatic ischemic stroke patients [13]. Multi-ethnic, genome-wide meta-analyses of dementia- and stroke-free subjects revealed that an SNP, rs9515201, at an intronic region of *COL4A2* is associated with WMH in community populations, as well as stroke patients. This SNP is in strong LD with three SNPs previously identified to be associated with sporadic ICH [73]. We recently confirmed the association of rs9515201 with WMH, particularly in a subgroup of extreme cases versus controls in an independent cohort with European ancestry [12]. Other previously identified WMH risk loci, such as rs3744028 (*TRIM65*) and rs1801133 (*MTHFR677* cytosine/thymine), have also been validated in this cohort. 

A large population-based GWAS from the UK Biobank further explored the role of common genetic variants contributing to cerebral microvascular health according to the measure of microstructural integrity of the white matter [76]. A significant genome-wide locus associated with both mean diffusivity (MD) (rs13164785; *p* = 3.7 × 10^−18^) and fractional anisotropy (FA) (rs67827860; *p* = 1.3 × 10^−14^) was identified at an intronic region of *VCAN*. This locus was nominally associated with WMH in the same study. *VCAN* encodes ECM proteoglycan versican, a versatile protein that plays a role in intercellular signaling and connecting cells with ECM [79]. 

In a transcriptome-wide association study (TWAS) [14] on WMH from the UK Biobank and other sources, key ECM proteins, such as *COL4A2*, *LOX*, *VCAN,* and *ADAMTSL4*, were associated with WMH or two other imaging traits (MD and FA), providing support for the hypothesis that the disruption of the cerebrovascular matrisome plays a central role in the pathogenesis of both monogenic (Mendelian vascular disease) and sporadic CSVD. In a TWAS on WMH (*n* = 50,970) of older individuals after accounting for modification/confounding by hypertension, some WMH risk loci were identified independent of blood pressure or other known vascular risk factors; two of these risk loci (*NID2* and *VCAN*), along with *COL4A2* and *EFEMP1*, implicate genes coding matrisome protein [80]. 

### 5.3. Genes Associated with Blood–Brain Barrier (BBB) Integrity May Contribute to the Risk for CSVD

A meta-analysis of GWAS with subsequent functional validation identified common variants near *FOXF2* associated with increased stroke susceptibility [17]. Seven of the eight known loci associated with risk for IS were replicated in this study, and a novel locus at 6p25 (rs12204590, near *FOXF2*) was identified to be associated with risk for all stroke (OR:1.08, 95%CI: 1.05–1.12, *p* = 1·48 × 10^−8^). The stroke risk allele of rs12204590 is also related to increased WMH. Consistently, young patients (aged 2–32 years) with segmental deletions of *FOXF2* have shown an extensive burden of WMH [17]. 

*FOXF2,* encoding a transcriptional factor, is expressed specifically in CNS pericytes and regulates pericyte differentiation and BBB development [81]. *FOXF2* knockout mouse embryos have shown developmental defects in the BBB, and deletion of *FOXF2* in adult mice results in cerebral infarction, reactive gliosis, and microhemorrhage. Pericytes play an important role in the regulation of BBB permeability, angiogenesis, clearance of cellular debris, immune cell entry, and cerebral blood flow [82]. 

Another gene that is expressed in CNS pericytes and as a part of integral components of the BBB is *FOXC1*. It encodes a forkhead box transcriptional factor, 225 kb, downstream of *FOXF2*. *FOXC1* is also expressed in brain vasculature and plays a role in pericyte function, such as vessel morphogenesis [83]. Patients with *FOXC1*-attributable Axenfeld–Rieger syndrome (mutation or copy number variation of *FOXC1*) have an increased WMH burden [84]. 

A meta-analysis of GWAS data in +/− 500 kb of *FOXC1* on 6p25 in 9361 individuals with brain MRI data from the CHARGE consortium identified WMH-associated SNPs (*p* = 0.0031–0.048, Bonferroni-corrected) located upstream of *FOXC1* [17]. These SNPs are eQTLs for *FOXC1*. 

### 5.4. Genetic Basis of Sporadic Cerebral Amyloid Angiopathy

Sporadic cerebral amyloid angiopathy (CAA) is characterized by progressive deposition of Aβ in the walls of cortical and leptomeningeal small arteries, resulting in vascular occlusion, rupture, and brain parenchymal damage [85,86].

Aβ, with peptides of 36–43 amino acids, is the main component of the amyloid plaques found in the brains of Alzheimer’s patients. These peptides originate from an amyloid precursor protein (APP), which is cleaved by β and r secretase to yield Aβ. The sporadic Aβ type of CAA is commonly found in the elderly and patients with Alzheimer’s disease. Aβ has been identified to cause Alzheimer’s disease, hereditary cerebral hemorrhage with amyloidosis [87], and CAA without symptomatic hemorrhage [85]. Aβ-induced toxicity includes the generation of reactive oxygen species, which trigger a signaling pathway to inflammation and apoptosis [88]. Recent studies showed that CAA-linked β-amyloid mutations (E22Q and D23N) promoted cerebral fibrin deposits via increased binding affinity for fibrinogen [89].

Hereditary cerebral hemorrhage with amyloidosis Dutch-type (HCHWA-D), an autosomal dominant disorder, is a hereditary Aβ type of CAA characterized by recurrent lobar cerebral hemorrhages; leukoencephalopathy has been reported to occur in middle age in Dutch families [90,91]. Although no neurofilament tangles or neurite plaques were observed, a point mutation was identified at codon 693 [92], leading to a substitution of Glu by Gln at position 22 in βAPP, a precursor protein of Aβ. Mutations in the nearby codon of βAPP were also found in a Dutch family with presenile dementia and CAA-related hemorrhage. In addition, familial Alzheimer’s disease caused by mutations in *PSEN1* and *PSEN2* genes frequently presents with severe CAA of the Aβ type [93].

The involvement of amyloid-ß precursor protein (APP) in the pathogenesis of stroke has not been emphasized as much as that of atherosclerosis. We also identified “the regulation of amyloid β formation” and “the regulation of APP catalytic pathway” as one of the top pathways specifically associated with IS beyond current etiological classification. Increased expression of APP and the production of Aß result in the formation of cerebral amyloid angiopathy (CAA) [94]. In our study [24,94], minor alleles with decreased risk for IS exhibited decreased expression of *APP* (i.e., rs138725707 as eQTL for *APP* with β = −0.11/−0.27 and *p* = 5.4 × 10^−6^/1.5 × 10^−7^ from eQTLGEN/Blueprint). APP and Aß production cause cerebral vascular cell death and enhance expression of matrix metalloproteinases and plasminogen activator proteolytic systems, leading to loss of vessel wall integrity and hemorrhage. Alternatively, elevated Aß levels and CAA cause vasoconstriction, thrombin production, platelet activation, fibrin deposition, and cerebral vessel dysfunction, contributing to cerebral ischemia.

Pathway-specific PRS derived from gene sets of APP and amyloid β formation is not only associated with IS but also post-IS long-term mortality [95]. Results of the subgroup analysis also highlighted several β amyloid peptide (Aβ)-related pathways associated with IS and post-IS long-term mortality solely in the elderly subgroup. Therefore, all the above findings link CAA, Aβ, apoptosis, inflammation, and fibrinolysis-related pathways identified in this study together. The contribution of common genetic variants from other amyloid proteins individually or together as PRS to the risk of sporadic CAA, IS, or post-IS mortality is still unknown and worthy of further investigation.

## 6. Pathway-Specific PRS Analysis for IS

Instead of creating a single PRS across the genome, the signal-to-noise ratio can be improved by reducing the noise contributed by irrelevant pathways. The latter can be achieved by limiting the hypotheses and generating PRS from gene sets. Pathway-specific PRS analyses may confirm known and identify novel etiologies related to IS and its subtypes. Advanced algorithms, such as transcriptome-wide association study (TWAS) [96], sequence kernel association test (SKAT) with optimal unified approach (SKATO) [97], and the regularization procedure accounting for LD via a reference panel (lassosum) [28], make this genotype–phenotype association converge at the gene and pathway level. This domain-knowledge-based approach can largely reduce the hypothesis space and heterogenicity, thus improving statistical power and generalizability. The effects estimated in large-scale GWAS serve continuously as building blocks for polygenic modeling. Prioritized genes or pathways could become potential drug targets or contribute as features in the predictive modeling of diseases and outcomes of interest.

### 6.1. Pathway-Specific PRS Construction

Pathway-specific PRS can be constructed using an array of algorithms, with the exploration of various modeling efforts still expanding. In the case of ischemic stroke, the PRS_avg_ derived from gene sets defined by the gene ontology (GO) biological process were calculated to test their association with IS under two MAF thresholds (MAF < 0.025 or <1) representing low-frequency common variants or all variants, respectively. GO pathways of Biological Processes and their related genes were defined by the Molecular Signatures Database (“msigdb_v7.0_GMTs/c5.bp.v7.0.symbols.gmt” from https://www.gsea-msigdb.org/gsea/msigdb/index.jsp). The gene sets can be drawn from curated pathway databases not limited to GO, e.g., subcellular location (e.g., matrixsome or synaptosome), functional annotation, predefined disease risk genes, coexpression network genes, drug target genes, protein–protein interaction databases (e.g., STRING or Reactome), or even self-defined gene sets based on the results of functional studies. In this study, we used a self-contained *p*-value to filter but not rank pathway-specific PRSs. However, for the purpose of gene-set enrichment analysis, a competitive *p*-value using the permutation approach is necessary to account for the size difference across pathways. Substantial computational resources are required, given the thousands of candidate pathways being tested.

Gene-set analyses illustrated the top pathways enriched for IS when the PRS was constructed based on each of the five MEGASTROKE summary statistics (color of the bars) stratified by index age for controls ≥ 69 years or 79 years (Figure 5), as shown in the y-axis and two levels of MAF for the included variants. Several known or novel pathways related to IS or its subtypes were identified. Gene set related to negative GO regulation of the (RNA) biosynthetic process, a downstream metabolic process, were ranked at the top using PRS_LAS_, PRS_CES_, and PR_SAS_. This suggests that selection may have occurred with respect to a variety of biological processes underlying metabolic processes in ways that differ among populations, such as people with cardiovascular diseases [34,98]. Here, we selected the “VEGF signaling pathway”, “regulation of APP catabolic process”, and “negative regulation of interleukin 2 production” as examples to highlight the specificity of these enriched pathways for the PRS constructed by the polygenic architecture of MEGASTROKE AS or AIS (Figure 5A). A VEGF-related gene network with variants contributed to the PRS was illustrated by String-db. With a similar approach, we also showed the pathways specific to SVS (purple bar in Figure 5B), CES (green bar in Figure 5C), and LAS (blue bar in Figure 5D). Neither “apoptotic process” nor “negative regulation of biosynthetic process” wasspecific to LAS; only the “cytokine production pathway” was specific to LAS (blue bar). The enriched pathways, such as “protein-lipid complex assembly” and “macrophage-derived foam cell differentiation”, confirm the pathology of atherosclerosis in the pathogenesis of IS and its subtypes. In addition, the phenotypic variation can be explained more by these relatively rare variants from genes (i.e., *ABCA1, PPARG, PRKCH*, and *APOB*), which were actively engaged in the process of atherosclerosis and became a drug target for stroke [99,100,101]. As one of several low-frequency variants associated with IS in *APOB*, rs1800480 is an eQTL for LDL cholesterol levels with the same direction for higher LDL levels (*p* = 1.6 × 10^−127^) [102]. High levels of apolipoprotein B/AI ratio are associated with intracranial atherosclerotic stenosis [103]. The results of the pathway enrichment analyses, as a proof-of-concept, confirmed several known mechanisms of atherothrombotic and lacunar stroke [104], which include atherosclerosis, the inflammatory system, lipid metabolism, endothelial function, thrombosis, and hemostasis, as identified through the candidate gene approach.

The VEGF signaling pathway was reported to be involved in the pathogenesis of stroke in two ways [105], with beneficial and deleterious effects. The beneficial effects includes collateral formation during development, reactive(reparative) angiogenesis toward oxygen depletion during the early stage of ischemic stroke, and profound neurogenesis. The deleterious effects comprise vascular leakage, as well as BBB breakdown. In this study, we found that risk alleles in VEGF gene networks showing increased risk for stroke have exhibit lower expression of these genes (i.e., rs78305106 as eQTL for FLT with β = −0.396 and *p* = 5.1 × 10^−32^ from eQTLGEN; rs34231037 as plasma pQTL for KDR with β = −1.15 and *p* = 1.0 × 10^−70^ from Sun et al. [106]). This could reflect the insufficiency of VEGF signaling as the etiology of IS in some patients.

Using pathway-specific gene sets to construct PRS could help to identify individual at high risk for a specific pathway and for the development of personalized management based on risk stratification, especially if a targeted therapy or lifestyle modification becomes available to manage the specific genetic risk. This process and modeling strategy are also interpretable, making them more trustworthy in a healthcare system setting with complex dynamics striving for transparency and confidence.

### 6.2. A Modified Paradigm of IS Risk Stratification beyond TOAST Subtyping

The primary goal of diagnostic stroke evaluation is to identify the underlying etiology so that targeted treatments can be designed and implemented to prevent a recurrence [2]. Several classification systems have managed to stratify stroke etiologies into discrete clinical, radiographic, and prognostic categories. Despite a decade of GWAS on IS and its subtypes, genetic evidence currently has only been considered under certain circumstances, in which prothrombotic abnormalities should be considered as a cause of stroke exclusively in patients with a history of unexplained thromboembolic events in young stroke patients who have no other explanations for their stroke [107,108,109]. There is an unmet need for the etiologic classification of strokes with multiple potential mechanisms into specific etiologic classes in the absence of evidence-based strategies, such as risk factors, family history, and medication, and to better quantify multiple competing causes in a given patient [110,111]. How genetic information from GWAS contributes to this etiologic classification of strokes and may assist in identifying the etiology of strokes of unknown origin, referred to as cryptogenic strokes, is still unclear. Mechanism-targeted treatments are not available for cryptogenic strokes, which represent 25% to 30% of IS, increasing the likelihood of have recurrent events. The quality of etiologic classification depends on the ability to generate homogenous subtypes with discrete outcomes (discriminative validity) and the clarity of classification rules to ensure utility in different settings with different investigators (reliability) [2]. It is necessary to further categorize IS using more homogenous groups stratified by risk factors, including PRS, and refine the current diagnostic system for subtyping. Whether PRS may augment the newer clinical classification systems (e.g., ASCO and CCS) should be determined, as these newer schemes may better stratify the stroke etiology, at least in some patients.

Based on the pioneer studies by consortia [16,74] and our PRS modeling and that of others [46,112], we propose a modified paradigm of IS risk stratification beyond TOAST subtyping to incorporate genetic information into the existing ideological classification system (Figure 6). According to the preliminary data [24], we simulated five lists of pathway-specific PRSs that are significantly associated with the corresponding clinical IS subtypes (TOAST) using the priors (regression coefficients) estimated based on the TOAST subgroup GWAS(MEGASTROKE). Subtype-specific pathways can be enriched by ranking the significance and the competitive *p*-value of the associations. To understand the relationships between subtypes, it is important to distinguish IS subtypes (unique) and to identify shared pathways between IS subtypes (shared) (see the vertical bar plot in Figure 6). The intersect pathways (distinct mode) across some or all clinically defined subtypes and the unique pathways toward each subtype could help to create new risk stratification procedures.

## 7. Pathway-Specific PRS Analysis of Post-IS Mortality

### 7.1. Pathway-Specific PRSs Augment Etiologic Subtyping of IS and Outcome Prediction

Risk stratification by integration of genetic and nongenetic risk factors and integration of multiple genetic architectures derived from stroke-related phenotypes to create so-called “metaPRS” could be an optimal direction for predictive modeling. Because overall risk for IS is determined by an interplay between genetic and clinical factors, a metaGRS was developed via a penalized regression model (Elastic Net) to integrate multiple sets of GWAS summary statistics on stroke or its modifiable clinical risk factors, which include hypertension, diabetes, dyslipidemia, BMI, and coronary artery disease (CAD) [112]. The hazard ratio of this metaGRS for IS doubles that of previous GRS in the UKB cohort for individuals with high metaGRS achieving currently recommended risk factor levels. However, this metaGRS approach remains insufficient to manage risk [112] because the genetic and phenotypic heterogeneity makes it difficult to stratify the risk based on a single integrated risk score, i.e., metaGRS. Genomic and phenotypic partitioning could provide an alternative solution for risk stratification at the subpopulation level. 

Exploring the genetic architecture in each disease subtype could augment clinical subtyping and help to better define or redefine subtypes. The discovery of pathway-specific PRS is expected to contribute to efforts to explore the novel pathways related to pathogenesis and validate therapeutic targets. This approach may assist in drug development by identifying new drugs with higher efficacy and by informing repositioning of existing drugs toward new disease indications. By constructing a pathway-dependent polygenic architecture underlying the known and novel pathogenesis, the predictive power could be significantly improved for at least a subset of patients harboring those genetic variants. 

Applying machine learning algorithms to feature selection and prediction by integration of pathway-specific PRSs into the disease or outcome predictive modeling [113,114,115] may enhance the predictive power. The latter is likely due to the fact that only a limited number of pathways are responsible for the pathogenesis. Evaluation of effect size of some pathway-specific PRS (e.g., β amyloid-related) in IS subtypes, as well as post-IS bleed events (e.g., ICH, CMB, and cSAH) and management, is of interest to determine the trajectory of IS [116], especially in younger patients, for whom improved management can lead to decades of improved quality of life. 

Post-IS mortality is multifactorial with known or unknown etiologies. Pathway-specific PRSs individually contribute to post-IS mortality, and together with clinical risk factors, better predict post-IS mortality. The risk of stroke death and stroke hospitalization in monozygotic compared with dizygotic co-twins is increased, with the heritability estimated as 0.32 and 0.17 respectively, suggesting that genetic liability contributes to post-stroke mortality [117]. We can prioritize some mortality-related PRSs among these IS-associated, pathway-specific PRS candidates [24], further demonstrating their independent predictability in an integrated prediction model, which is superior to that of the base model (clinical risk factors only). A total of 15 PRSs formed by SNPs with MAF < 1 and 16 PRSs formed by SNPs with MAF < 0.025 from 31 unique gene sets were treated as disease-relevant, pathway-specific PRS (p_raw_ < 0.1). Kaplan–Meier analyses (Figure 7) showed that all 31 PRS candidates significantly distinguished the high- and low-risk groups with p_raw_ < 0.05.

### 7.2. Improved Predictability of Pathway-Specific PRS for Post-IS Mortality Using an Integrated Cox Proportional Hazards Model

Improved predictability can be achieved by better interrogating the data and by using methodologies that are carefully aligned with the data characteristics. Owing to the hierarchical nature of GO biological process terms, multicollinearity of PRSs is common. There are also extensive correlations within or between PRSs and clinical risk factors. All these factors can inflate the regression coefficients of predictive variables in the multivariate regression model. An *L1* penalization technique (LASSO regression) can handle this situation by forcing some regression coefficient estimates to be exactly zero, thus achieving variable selection while shrinking the remaining coefficients toward zero to avoid the overfitting and overestimation caused by data-driven model selection. The least absolute shrinkage and selection operator (LASSO) method [118] in the multivariate Coxph model was applied for feature selection of prognostic pathway-specific PRSs [95]. A prediction model including an additional 16 disease-associated pathway-specific PRSs outperformed the base model (8 clinical risk factors), as demonstrated by a higher concordance index (0.754, 95% CI: 0.693–0.814 versus 0.729, 95% CI: 0.676–0.782, respectively) in the holdout sample (*p* < 0.001 for the median improvement). Compared to the base model, the integrated PRS prediction model differentiated not only the high-risk from the intermediate-risk (*p* = 0.006) but also the intermediate-risk from the low-risk (*p* = 0.001) (Figure 8). Thee PRS derived from GO negative regulation of endothelial apoptotic pathway was the independent predictor for 3-year post-IS mortality (HR = 1.203) [95].

### 7.3. Validation of Exiting Etiologies and Drug-Targeting Pathways

The identified pathway-specific PRSs highlight the known pathogenesis of IS or post-IS mortality [95]. These pathways include, (1) amyloid β formation in cerebral small vascular disease; (2) endothelial apoptosis [119,120,121], inflammation (interleukin 1 [122,123,124,125], and tumor necrosis factor (TNF) [126,127]) in atherosclerosis; (3) serotonin in platelet aggregation [128,129,130,131] and vascular remodeling [132,133,134,135] and neuroplasticity in post-stroke recovery [136,137,138,139,140]; (4) obesity paradoxical in post-IS mortality [141,142]; and (5) coagulation and fibrinolysis in stroke and recurrence.

The results from the correlation between PRSs and modifiable clinical risk factors indicate that several pathways might also contribute to modifiable clinical risk factors, suggesting horizontal or vertical pleiotropisms. One of the most clinically relevant findings is the association of multiple pathway-specific PRSs with AFib with the same direction for disease risk and mortality risk, particularly in the elderly subgroup. This includes pathways related to fibrinolysis, APP and amyloid β formation, T cell differentiation, glomerular basement membrane development, positive regulation of membrane depolarization, and response to TNF.

## 8. Future Perspectives

Development and evaluation of novel methods for PRS construction will continue to be a driving force to move the field of statistical and epidemiological genetics forward, particularly for populations with mixed ancestry or underrepresented populations. Determining which methods can provide better parameter estimation in a polygenic framework is still an area of ongoing research. Advances in phenotyping using diagnostic codes from electronic health records have revolutionized phenome-wide association studies (PheWAS) and improved the quality of feature selection in multivariate predictive modeling of outcomes of interest. EHR-based phenotyping has also expanded the horizon to increase the sample size of cases and controls, in addition to providing venues to include multiple types and ratios of controls. Multiple areas of genetic research will benefit from these advances, including identifying more (endo)phenotype-specific loci, exploring the genetic correlation between risk factors and traits, assessing the causal contribution of novel biological pathways to disease risk through MR approaches, and exploring the disease trajectory for PRS-based risk-stratified individuals [143]. MR will gain popularity as a way to access the causal effect of exposure on outcome of interests in observational cohorts. Functional annotation of genetic variants according to curated QTL databases (eQTLs, methylation QTL, and plasma QTL) [144,145] will contribute to understanding of regulatory variants and prioritization of potential causal variants for PRS construction and integration into risk models.

### 8.1. The Utility of PRS in Mixed or Underrepresented Populations

PRS has to be applicable to all patients in the population, regardless of ethnicity, to ensure health equity in the distribution of healthcare resource [146]. There is an urgent need to generalize polygenic scores to patients with non-European ancestry. Due to differences in variant frequencies and LD patterns between populations with different ancestry, reduced predictive power in non-European ancestry samples is anticipated, particularly in patients with African ancestry. [147] In addition, data resources for non-European populations are currently inadequate, resulting in the rationale for large-scale GWAS in diverse human populations. Having realized the full and equitable potential of PRS, we should promote genetic studies on underserved populations [148] and the development of novel strategies to better estimate the effects on minorities (especially when it is impractical to reach a sufficient sample size due to a small population size). The American Heart Association recently made a scientific statement to encourage genetic/genomic research on marginalized ethnic groups to avoid historical harm in biomedical research due to the neglect of such populations [149].

The predictability of PRSs is expected to be augmented in a more homogenous population as the effect size and their interaction with environmental factors would be more specific. This could decrease the complexity of the model with a better bias–variance tradeoff when the discovery and validation cohorts are presumably sampled from the same population. Whether to create a universal PRS framework applied to a population with diverse ancestral backgrounds or to create multiple ancestry-specific PRS applied to matched subpopulations is the subject of ongoing debate.

### 8.2. The Challenges of Integrating PRS into Clinical Decision Support Systems and Risk Stratification Procedures

The phenotype variation for cardiometabolic traits can be partially explained by genetic variation from common variants with moderate inheritability [150]. As aging is an accelerator of cardiometabolic traits, late-onset phenotypes are less likely to be caused by genetic variation derived from common variants when compared to early-onset counterparts [150]. However, this may not be the case for ultra-rare variants identified from disease risk genes [5]. The varied expressivity of rare variant carriers could be due to an interplay between complex (polygenic) and Mendelian (monogenic) genetics for IS [151,152]. PRSs derived from common variants could modify the outcome of patients with a monogenic disease [151,153]. To determine the genetic burden at the individual level, it is crucial to develop genetic risk scores formed by both common and rare variants and variants beyond single-nucleotide polymorphisms.

How to integrate PRSs derived from informative (endo)phenotypes with high inheritability into the prediction model by advanced machine learning approach has yet to be fully elucidated. The complexity and multifactorial nature of IS and subtype-specific (poly)genetic markers should be deeply considered with respect to the development of an implementation approach for outcome prediction, as the effect size of such biomarkers could vary. It is crucial to apply suitable ML models to account for indirect and nonlinear associations between predictive variables (genetic or nongenetic), as well as the outcome of interest. Finally, the real effect size for PRS could vary across subpopulations defined by demographic features (sex, age, race, and ethnicity) and clinical subtypes. Whether PRSs can be treated as independent variables in multivariate models should be evaluated in subgroup analyses.

## 9. Conclusions

In this paper, we discussed the polygenic nature of IS and emphasized the role of PRS in risk stratification for disease/outcome prediction and personalized management of IS. Polygenic risk for cardiovascular disease may also account for the risk of sporadic IS. PRS may augment IS subtyping. We introduced a pathway-specific PRS analysis and demonstrated its utility in confirming known and identifying novel etiologies of IS. Some of these specific PRSs (e.g., derived from the endothelial cell apoptosis pathway) individually contribute to post-IS mortality, and together with clinical risk factors, better predicted post-IS mortality. Statistical models and machine learning algorithms have contributed significantly to the advancement in this field and will continue to drive innovation for genome-based healthcare decision making striving toward innovation, equity, and transparency.

## Figures and Tables

**Figure 1 jcm-11-05980-f001:**
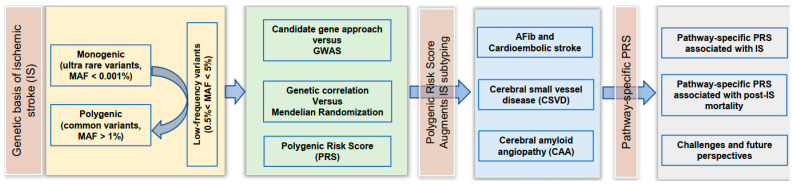
Flow chart summarizing the topics discussed in this review article.

**Figure 2 jcm-11-05980-f002:**
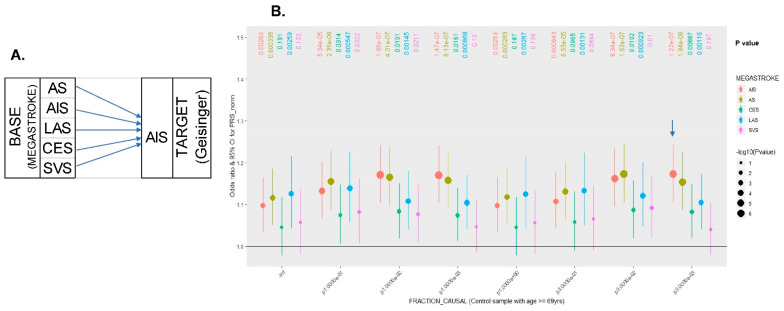
LDpred (nonuniform shrunk) to determine which fraction of causal variants provides the best prediction of the phenotype (ischemic stroke). (**A**) Layout of the PRS construction BASE files collected from the MEGASTROKE consortium used to estimate the effect size for each variant; the TARGET file is referred to as the genotyping file from the Geisinger Stroke Registry (GNSIS). (**B**) PRS calculated by LDpred based on the non-infinitesimal distribution (Gaussian mixture prior) of effect sizes compared to that based on infinitesimal distribution of effect sizes. PRS was normalized and, a logistic regression adjusted according to the covariates, including sex, PC_1–5_ was performed to determine the association with IS. Because the causal variant fraction (ρ) is unknown for any given disease, a range of ρ values (i.e., 1, 0.3, 0.1, 0.03, 0.01, 0.003, and 0.001) was used. This process helped to determine which fraction of causal variants provided the best prediction of phenotypes. The seven candidate LDpred scores vary with respect to the tuning parameter (ρ) which is the proportion of variants assumed to be causal. The PRS generated from MEGASTROKE AS and AIS predicted Geisinger achieved better IS prediction results with larger ORs (i.e., a fraction of causal variants at 0.003 resulted in the largest ORs using AIS summary statistics, as indicated by the arrow) than the PRS generated from MEGASTROKE subtypes, such as LAS, SVS, and CES. Overall, these LDpred scores based on the non-infinitesimal distribution of effect sizes were larger than those obtained using infinitesimal distributions.

**Figure 3 jcm-11-05980-f003:**
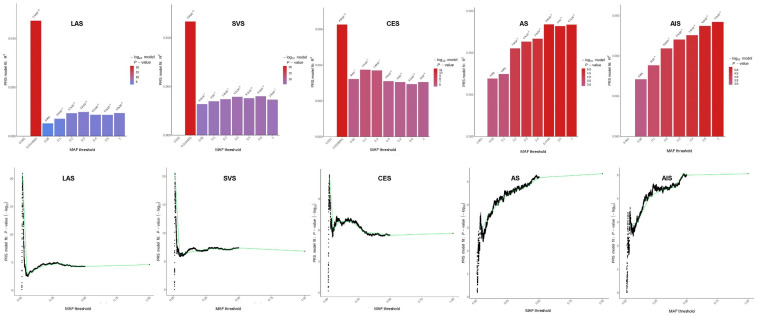
PRS derived from common genetic variants with low minor allele frequency (MAF < 0.025) provided the best-fit modeling for ischemic stroke when PRS was constructed based on the summary statistics of TOAST subtypes, such as LAS, SVS, and CES. PRS was constructed by PRSice-2. High-resolution plots were used to select the consistent cut-off value for the MAF threshold for PRS construction and the gene-set analyses. Discovery cases (*n* = 1184) vs. controls (*n* = 19,806). This figure adapted from Li J (2021) [24] with CC BY-NC-ND.

**Figure 4 jcm-11-05980-f004:**
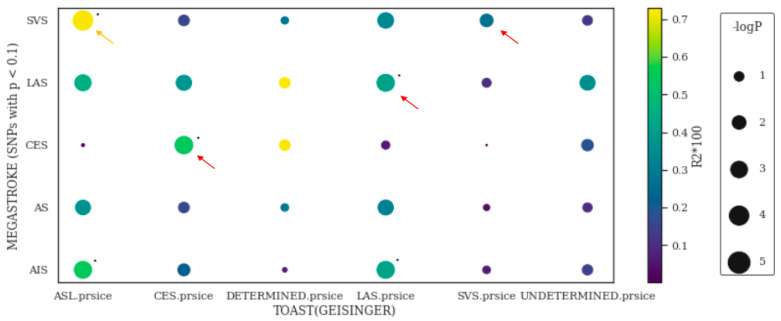
The PRS derived from MEGASTROKE subtypes was mostly associated with the corresponding Geisinger TOAST subtypes. The dot plot demonstrates the association of PRS derived from MEGASTROKE on Geisinger TOAST subtypes when using base *p* < 0.1 as an example. The association between PRS and stroke subphenotypes was tested by logistic regression (Phenotype ~ PRS_avg_ + Sex + PC_1–5_). The PRSs derived by the MEGASTROKE consortium (y-axis) was calculated by PRSice-2 to determine their association with the TOAST subtypes of Geisinger ischemic stroke patients (*x*-axis). Nagelkerke pseudo-*R*^2^ (color of dots) and significant levels (size of dots) were calculated by PRSice-2 with clumping and thresholding (using SNPs with base *p* < 0.1 as an example). “*” represents the significance of the association surviving Bonferroni correction given 30 paralleled testing (P_unadjusted_ < 0.0017)”. We excluded any cases with a recurrent stroke of different TOAST subtypes. CES, cardiac embolism; DETERMINED, stroke of other determined etiology; LAS, large artery stroke; SVS, small vessel stroke; UNDETERMINED, stroke of undetermined etiology; ASL, a synthesized TOAST subtype, which represents a combination of Acute SVS (*n* = 79) and LAS (*n* = 124); AS, any stroke; AIS, any ischemic stroke. This figure adapted from Li J (2021) [24] with CC BY-NC-ND.

**Figure 5 jcm-11-05980-f005:**
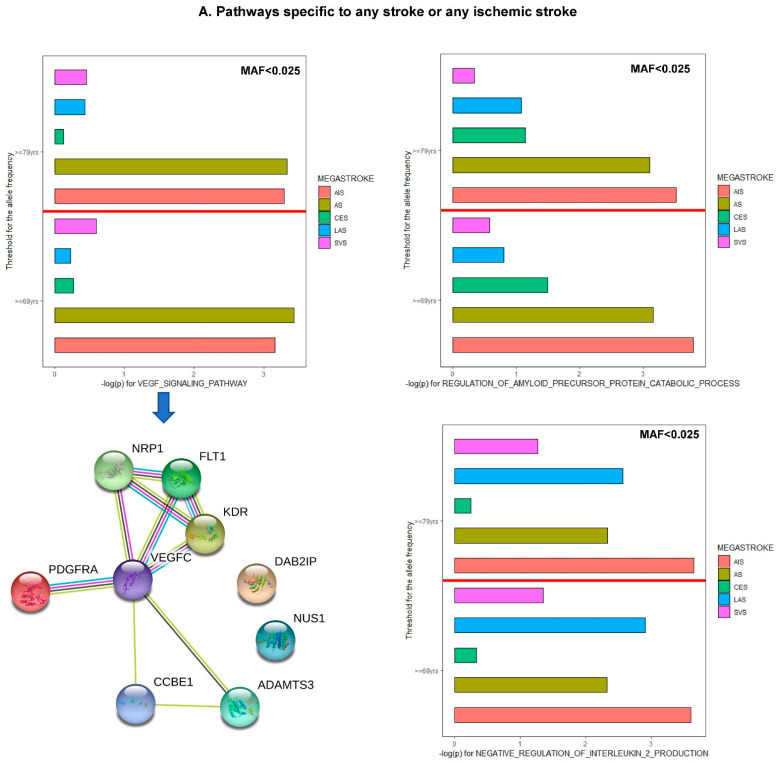
Pathways specific to any stroke or any ischemic stroke. Gene-set analyses illustrate the top pathways enriched for ischemic stroke when the PRS was constructed based on each of the five summary statistics of MEGASTROKE (color of the bars) stratified by controls with index age ≥ 69yrs or 79 yrs, as shown in the *y*-axis. Here, we selected the VEGF signaling pathway, regulation of amyloid precursor protein catabolic process, and negative regulation of interleukin 2 production as examples to show the specificity of these pathways significantly enriched only for the PRS constructed by the polygenic architecture of MEGASTROKE AS or AIS. PRS was constructed under two levels of MAF thresholding. Here, we only chose MAF ≤ 0.025 as an example to show that these pathways were the top pathways under this condition. All the raw *p* values in the x-axis demonstrate the significance of the enriched signals against the background without correction for multiple testing. The gene network that harbors those genetic risks with a moderate effects size for ischemic stroke (*p* < 0.1) were illustrated using String-db. Here, we chose genes included in the VEGF signaling pathway as an example.

**Figure 6 jcm-11-05980-f006:**
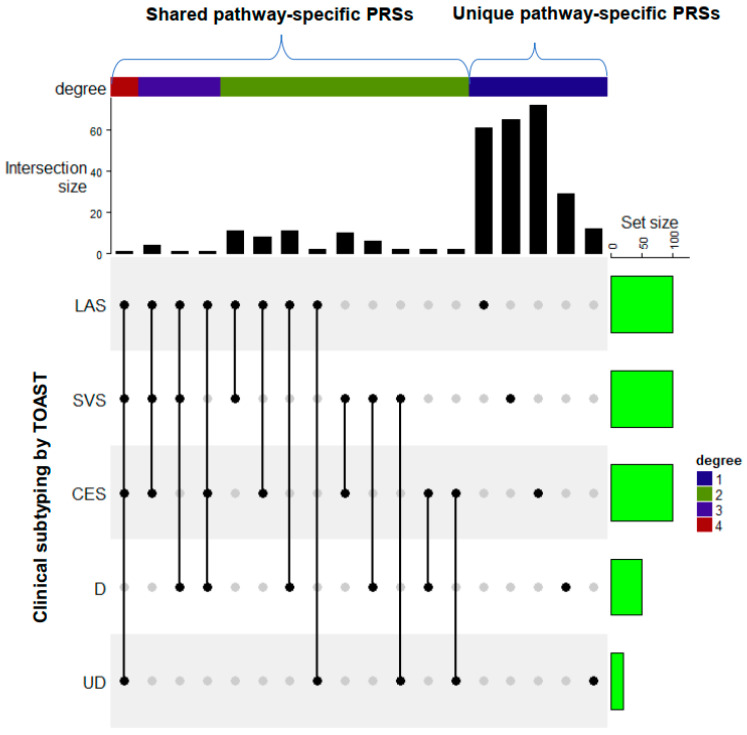
A modified paradigm of IS risk stratification beyond TOAST subtyping using simulated data. Based on the pilot study, we assume a total of 700 valid pathway-specific PRSs are available for a subtype-specific association study. The horizontal bar plot represents the size of PRSs with significant association with the corresponding TOAST subtypes. TOAST subtypes include (1) cardiac embolism (CES), (2) stroke of other determined etiology (DETERMINED (D)), (3) large artery stroke (LAS), (4) small vessel stroke (SVS), and (5) stroke of undetermined etiology (UNDETERMINED (UD)). We assume 100 out of 700 pathway-specific PRSs showed at least minimal significance for LAS, SVS, and CES. However, this set size was assumed to be dramatically decreased for D and UD due to small sample size and less heritability for U or advanced heterogenicity for D. The vertical bar plot represents the number of pathway-specific PRSs shared between at least two TOAST subtypes or unique to individual subtypes. The intersect pathways (distinct mode) across some or all clinically defined subtypes and the unique pathways toward each subtype could help to create new risk stratification procedures.

**Figure 7 jcm-11-05980-f007:**
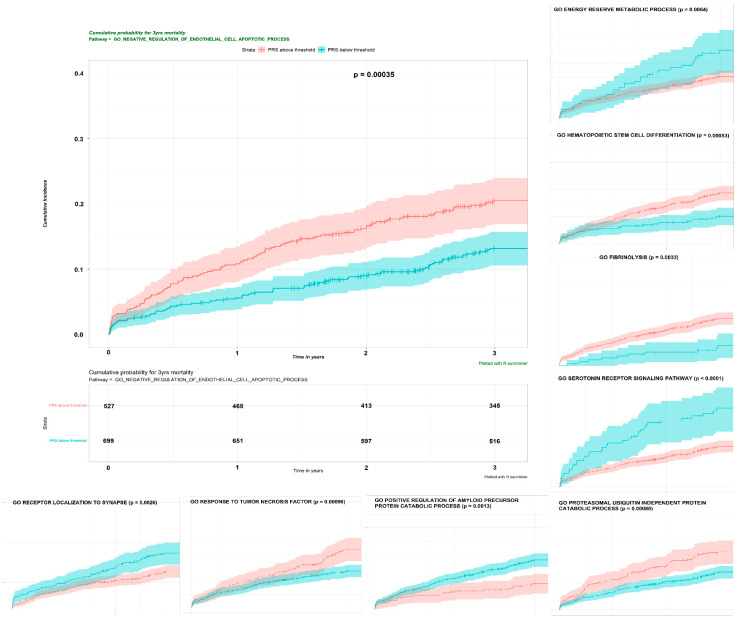
Kaplan–Meier plot of the two groups created by the cut point for PRS in the training dataset. All the pathway-specific PRSs for 3-year mortality identified by univariate Coxph were dichotomized by the corresponding cut point, and a Kaplan–Meier analysis was conducted for each binary PRS. A total of 31 PRS candidates significantly distinguished the high- and low-risk groups with p_raw_ < 0.05. Only the results from nine candidate PRSs are presented here due to space limits. We simulated the null distribution using the conditional *Monte Carlo* method and compared it with the exact distribution for the log-rank statistic to determine the lower bound of the *p*-value for each pathway-specific PRS. The *p*-value derived from the log-rank test is labeled. This figure adapted from Li J (2022) [95] with CC BY 4.0.

**Figure 8 jcm-11-05980-f008:**
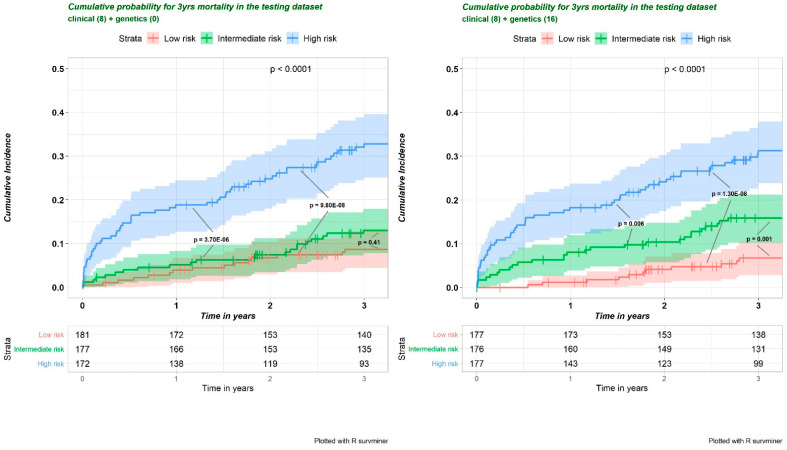
Kaplan–Meier analysis of post-IS cumulative probability for 3-year mortality in the testing dataset. Assuming three subgroups with different survival probability in the testing dataset to determine the effect size of each feature included in the multivariate Cox proportional hazards regression model. The *p*-value derived from the log-rank test is labeled. For all analyses, *p* < 0.05 was considered statistically significant. For all post hoc pairwise tests, *p* values were adjusted according to the Benjamin–Hochberg procedure. The number at risk is listed in the table. This figure adapted from Li J (2022) [95] with CC BY 4.0.

**Table 1 jcm-11-05980-t001:** CSVD-associated SNPs identified in genome-wide association studies.

SNP	Locus	Gene	Phenotype	*p*-Value	Reference
rs2230500	14q22-q23	*PRKCH*	Lacunar infarct	5.1 × 10^−7^	[67]
rs2230500	14q22-q23	*PRKCH*	ICH	5.4 × 10^−3^	[68]
rs4646994	17q23	*ACE*	WMH	NA	[69]
rs1055129	17q25	*TRIM47*	WMH	4.1 × 10^−8^	[70]
rs3744028	17q25	*TRIM65*	WMH	4.0 × 10^−9^	
rs4646994	17q23	*ACE*	Lacunar infarct	6.0 × 10^−3^	[36]
rs2984613	1q22	*PMF1 SLC25A44*	Non-lobar ICH	1.6 × 10^−8^	[71]
rs72848980	17q25.1	*NEURL1*	WMH	2.7 × 10^−19^	[72]
rs7894407	10q24.33	*PDCD11*	WMH	2.7 × 10^−19^	
rs12357919, rs7909791	10q24	*SH3PXD2A*	WMH	1.6 × 10^−9^	
rs78857879	2p16.1	*EFEMP1*	WMH	1.5 × 10^−8^	[72]
rs11679640	2p21	*HAAO*	WMH	4.4 × 10^−8^	
rs72934505	2q33.2	*NBEAL1*	WMH	2.2 × 10^−8^	[73]
rs941898	14q32.2	*EVL*	WMH	4.0 × 10^−8^	
rs962888	17q21.31	*C1QL1*	WMH	1.1 × 10^−8^	
rs9515201	13q34	*COL4A2*	WMH	6.9 × 10^−9^	
rs10744777	12	*ALDH2*	Small artery stroke	2.92 × 10^−9^	[74]
rs12204590	6p25	*FOXF2*	WMH	2.17 × 10^−6^	[17]
rs12445022	16q24	*ZCCHC14*	WMH	5.3 × 10^−5^	[75]
rs13164785, rs67827860	5q14.3	*VCAN*	MD, FA	3.7 × 10^−18^ 1.3 × 10^−14^	[76]
rs275350	6q25.1	*PLEKHG1*	WMH	1.6 × 10^−8^	[77]

Abbreviation: ICH, intracerebral hemorrhage; WMH, white matter hyperintensity; MD and FA, mean diffusivity and fractional anisotropy, respectively, two common DTI (diffusion tensor imaging) measures.

## Data Availability

Not applicable.

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
