# Peer review of "Dissecting Polygenic Etiology of Ischemic Stroke in the Era of Precision Medicine"

_jcm, 2022, doi:10.3390/jcm11205980_

Round 1

Reviewer 1 Report

This is a review article discussing about dissecting polygenic etiology of ischemic stroke in the era of precision medicine. This is a well-written article with comprehensive review of cerebral small vessel disease risk genes. The authors summarize the genetic basis of ischemic stroke and emphasize the application of methodologies and algorithms used when constructing polygenic risk scores and integrating genetics into risk models. This review article provides important information to the readers.

A major problem observed in this article with PDF format is that Figures are not found in the manuscript despite of the figure legends. Was it a technical problem with the website download system or is the manuscript preparation incomplete? Without the figures, the reviewers may not be able to understand the main parts of this review article.

Author Response

Thank the reviewer’s comment.

We are sorry to hear that the reviewer cannot see the figures. This could be a technique problem beyond our control. Please review the manuscript in the latest pdf format with the figures and tables integrated by MDPI.

Reviewer 2 Report

Great job!

I enjoyed reading this article. This is a comprehensive review of the polygenic etiology of ischemic stroke.

I have no objections to the text itself. The sections on stroke etiology and genetic influence are well written. The part about forecasting models in this area is not that familiar to me, so I may not have the best competence to give my judgment on that part.

Perhaps, considering the comprehensiveness, it could be divided into two parts.

Author Response

Thank the reviewer’s comment.

Reviewer 3 Report

The authors discuss and analyse in detail the genetic bases of ischemic stroke with particular attention to applying statistical methods, machine learning and algorithms to risk models.

The manuscript is very well written, it is clear and very exhaustive with excellent figures which help the reader to follow the review.

I have a comment which I hope will help the authors to improve with future research.

To stratify the stroke etiology the authors used the most common classification scale: the TOAST classification which divided the stroke etiology into 5 big categories. While on the one hand, the TOAST classification is practical and useful for clinicians, on the other hand, it does not take into consideration patients with multiple causes or patients with other specific or non-specific causes of stroke (such as artery dissection, patent foramen ovale, non-stenotic atherosclerosis, atherosclerosis of the aortic arch ..), not to mention the ESUS (embolic stroke of undetermined sources). There are more complex but complete classification systems (ASCOD, CCS) that may help better stratify the stroke etiology. I suggest the authors take them into consideration for future ideas.

Author Response

Thank the reviewer’s comment. The reviewer raises an important point of the current classification of ischemic stroke. Several classification systems have managed to stratify stroke etiologies into categories with discrete clinical, radiographic, and prognostic categories. Several studies have compared the TOAST to the newer stroke classification schemes such as the Causative Classification System (CCS) and A-S-C-O (A for atherosclerosis, S for small vessel disease, C for cardiac source, O for other cause) Phenotypic System (ASCO). The take-home-messages are the following. 1. Both CCS and ASCO schemes showed good-to-excellent agreement with TOAST, suggesting the major etiologic subtypes have distinct characteristics irrespective of the classification system; 2. CCS and ASCO had specific characteristics compared with TOAST for subtype assignment and data retention (Marnane M, et al. Stroke 2010;41:1579-1586). CCS generates discrete etiologic categories than either TOAST or ASCO (JAMA Neurol. 2017;74(4):419-426). 3. For GWAS, no matter which classification scheme (CCS, or TOAST) being used, each case has been categorized according to five ischemic stroke subtypes: cardioembolic, large artery atherosclerosis, small artery occlusion, undetermined, and other according to the NINDS stoke Genetics Network (Pulit SL, et al. Lancet Neurol 2016; 15(2): 174-184).

As the reviewer suggests, we have included the following sentences into the section “6.2. A modified paradigm of IS risk stratification beyond TOAST subtyping”.

Despite a decade of GWAS on IS and its subtypes, genetic evidence currently has only been considered under certain circumstances, …”.

“It is necessary to further categorize IS using more homogenous groups stratified by risk factors including PRS and refine the current diagnostic system for subtyping. Whether PRS may augment the newer classification systems (ASCO, CCS) should be determined as these newer schemes may better stratify the stroke etiology at least in some patients”. See line 628-630.
